# Postpartum Diet Quality: A Cross-Sectional Analysis from the Australian Longitudinal Study on Women’s Health

**DOI:** 10.3390/jcm9020446

**Published:** 2020-02-06

**Authors:** Julie C. Martin, Anju E. Joham, Gita D. Mishra, Allison M. Hodge, Lisa J. Moran, Cheryce L. Harrison

**Affiliations:** 1Monash Centre for Health Research and Implementation (MCHRI), School of Public Health and Preventative Medicine, Monash University, Melbourne 3000, Australia; jcmar8@student.monash.edu (J.C.M.); anju.joham@monash.edu (A.E.J.); lisa.moran@monash.edu (L.J.M.); 2Diabetes and Vascular Medicine Unit, Monash Health, Clayton 3168, Australia; 3Centre for Longitudinal and Life Course Research, School of Public Health, University of Queensland, Brisbane 4000, Australia; g.mishra@sph.uq.edu.au; 4Cancer Epidemiology Division, Cancer Council Victoria, Melbourne 3000, Australia; Allison.Hodge@cancervic.org.au; 5Centre for Epidemiology and Biostatistics, Melbourne School of Population and Global Health, University of Melbourne, Melbourne 3000, Australia

**Keywords:** diet quality, nutrition, obesity, prevention, postpartum, reproductive, women

## Abstract

Reproductive-aged women are at high risk of developing obesity, and diet quality is a potential modifiable risk factor. There is limited research exploring diet quality and its association with time since childbirth. Using data from the Australian Longitudinal Study on Women’s Health (ALSWH) survey 5 (2009) of women born between 1973–1978, who reported having previously given birth, we investigated the association between time since childbirth and diet quality, and differences in energy, macronutrients, micronutrient intake, and diet quality assessed by the dietary guideline index (DGI) in women stratified by time from last childbirth, early (0–6 months; *n* = 558) and late (7–12 months; *n* = 547), and all other women with children (>12 months post childbirth *n* = 3434). From this cohort, 8200 participants were eligible, of which 4539 participants completed a food frequency questionnaire (FFQ) and were included in this analysis. Overall, diet quality was higher in early and late postpartum women (mean DGI score 89.8 (SD 10.5) and mean DGI score 90.0 (SD 10.2), respectively) compared to all other women with children (>12 months post childbirth), mean DGI score 85.2 (SD 11.7), *p* < 0.001. Factors positively associated with diet quality included higher education, physical activity, health provider support, and vitamin and/or mineral supplement use. Conversely, increasing time from childbirth (>12 months), smoking compared with non-smoking and medium income level compared with no income was negatively associated with diet quality. A lower diet quality in women greater than 12 months post childbirth may be reflective of increased pressures, balancing childrearing and return to work responsibilities. This highlights the need to support women beyond the postpartum period to improve modifiable factors associated with weight gain, including diet quality, to optimize health and reduce chronic disease risk.

## 1. Introduction

Reproductive-aged women are a susceptible population for obesity development [1]. Over 50% of women are overweight or of obese preconception [2,3,4], and, once pregnant, 50%–60% of women gain more than the International Institute of Medicine gestational weight gain (GWG) recommendations [5]. Excessive GWG increases the risk of pregnancy and birthing complications [6], infant mortality [7], and maternal cardiovascular risk factors [8]. Furthermore, excess GWG drives postpartum weight retention (PPWR), increasing risk in subsequent pregnancies and fueling maternal obesity development associated with chronic diseases long-term [9,10]. 

Diet quality is a modifiable risk factor for obesity prevention and is inversely associated with weight gain [11,12,13,14,15], waist circumference [16,17], body mass index (BMI) [12,17,18], and cardio metabolic risk factors [12]. Diet quality assessed using scores, including the dietary guideline index (DGI) [19], provides an overview of diet in its entirety, including the assessment of dietary patterns, nutrient intake, and compliance with national dietary guidelines. A higher diet quality score implies increased consumption of a greater amount and variety of healthy foods and less consumption of energy-dense, nutrient-poor foods [19]. To date, associations between diet quality and weight change postpartum have not been explored; however, a higher total energy intake has been reported as an inverse predictor of weight loss postpartum [20]. 

Previous research has shown that diet quality may be suboptimal across reproductive life stages, specifically in pregnancy [21], postpartum [12,21,22], and in young mothers [23,24], with women failing to meet dietary recommendations for fruit and vegetables, wholegrains and legumes, lean protein foods, oils, dairy foods, saturated fat, added sugars, and sodium. However, there is limited research assessing diet quality across postpartum time periods [21] and little is known about predictors of poorer dietary intake, including demographic and lifestyle factors postpartum. Further, there is limited literature assessing population-based cohorts, with previous research focusing on specific populations, such as multi-ethnic, low income, and overweight and obese postpartum women [12,22,23,24,25]. Therefore, we aimed to investigate the association between time since childbirth and diet quality and the differences in diet quality, stratified by time from childbirth, including early postpartum (0–6 months) and late postpartum (7–12 months) and all other women with children (>12 months post childbirth) within a population-based cohort from ALSWH. Our secondary aim is to look at overall predictors of diet quality in women with children in order to target women at risk of poorer diet quality, as one strategy for obesity prevention in this vulnerable group at high risk for poor dietary intake and weight gain.

## 2. Experimental Section

### 2.1. Study Population 

This study is based on data from the ALSWH, a prospective cohort study of women’s physical and mental health, psychosocial aspects of health (such as socio-demographic and lifestyle factors), and use of health services [26]. In 1996, three cohorts of young (born 1973–1978, aged 18–23 years; *n* = 14,247), middle-aged (born 1946–1951, aged 45–50 years; *n* = 13,715), and older (born 1921–1926, aged 70–75 years; *n* = 12,432) women were recruited [26]. Women were randomly selected from the national health insurance scheme (Medicare) database, which includes women who are permanent residents of Australia, with national recruitment and intentional over-sampling from rural and remote areas [27]. The women completed a mailed survey every three years [27]. Further details of the methods and characteristics of the ALSWH have been reported elsewhere [26,27,28,29]. The recruited population of women has previously been shown to be reasonably representative of the general population, although ALSWH participants were slightly more likely to be Australian born and to have a post school qualification when first recruited in 1996 [28,30]. All subjects gave their informed consent for inclusion before they participated in the study. The study was conducted in accordance with the Declaration of Helsinki, and the protocol was approved by The Human Research Ethics Committees of the University of Newcastle and the University of Queensland. 

### 2.2. Participants 

This study uses data from the cohort of young women who completed survey 5 of the ALSWH (2009, *n* = 8200), because women in this age group (aged 31–36 years in 2009) had the highest age-specific fertility level compared to all other age groups [31]. From this total, we excluded those with an incomplete food frequency questionnaire (FFQ) (>10% of items missing responses; *n* = 86) or if their reported daily energy intake was >14,700 kj/day (3513.4 Kcal/day) (*n* = 94) or <2100 kj/day (501.9 Kcal/day) (*n* = 20) [32,33]. Due to differing dietary requirements compared with the general population, we excluded pregnant women (*n* = 804) and women who reported having a gastric band (*n* = 48). As the primary aim was to investigate how diet quality varied with time since childbirth, those that reported having no children (*n* = 2633) were excluded. Participants with an unidentifiable postpartum period (*n* = 5) were also excluded. After applying all exclusions, 4539 women who completed the FFQ within survey 5 were included for analysis, which were further categorized as early postpartum (0–6 months since birth of the last child) *n* = 558, late postpartum (7–12 months since the birth of the last child) *n* = 547, and women >12 month post childbirth *n* = 3434. 

### 2.3. Dietary Intake 

Self-reported dietary intake data was collected using the Dietary Questionnaire for Epidemiological Studies (DQES) Version 2, a semi quantitative FFQ, developed by The Cancer Council of Victoria and previously validated in young Australian women [34]. The DQES version 2 collects information regarding the consumption frequency of 80 food and beverage items, including cereal foods, dairy products, meats and fish, fruit and vegetables, and discretionary foods and beverages, including sweets or savory snacks and alcoholic beverages [35]. Women were asked to answer questions regarding their usual dietary intake over the last 12 months and indicate frequency of consumption from never, monthly, weekly, daily, and up to three or more times per day [35]. The FFQ also contained pictures of meals for participants to indicate their usual portion size, which also contributed to the final dietary analysis [35]. Nutrient intake was calculated using Australian food composition data [36].

### 2.4. Outcome: Diet Quality

Diet quality was measured by the DGI as an a priori scoring method validated for use in the Australian adult population [19]. The DGI reflects compliance with dietary guidelines for Australian adults and comprises dietary indicators of vegetables and legumes, fruit, breads and cereals, meat and alternatives, dairy, and extra foods, defined as non-essential foods and beverages that contain alcohol or high levels of saturated fat, sugar, and salt [19]. Due to the FFQ not providing specific information to fulfil the requirements of salt use, fluid intake, and saturated fat components, which specified the trimming of fat from meat, these components were excluded [19]. Each component was scored from 0 to 10, with 10 indicating an optimal intake. The total score was the sum of 13 indicators with the DGI having a possible range of 0–130, with a higher score indicating better compliance with the dietary guidelines [19].

### 2.5. Independent Variable: Time Since the Last Childbirth

Each woman’s postpartum period was calculated by using the survey date minus the last child’s date of birth. Women were categorized as early (0–6 months), late (7–12 months) postpartum, and all other women with children (>12 months after the birth of the last child), consistent with standard definitions [37,38]. 

### 2.6. Confounders

#### 2.6.1. Physical Activity

Self-reported physical activity was based on frequency and duration of a variety of activities performed in the last week for 10 min or more. These included moderate and vigorous leisure time activity, as well as active transportation-related activity, as previously reported [39]. Physical activity was calculated as the sum of the products of total weekly minutes in each of the three categories of physical activity and the metabolic equivalent value (MetMin) assigned to each category: (walking minutes × 3.33 MetMin) + (moderate intensity physical activity minutes × 3.33 MetMin) + (vigorous intensity physical activity minutes × 6.66 MetMin) [40,41]. Outliers were truncated at 28 h per week for total physical activity [32,33].

#### 2.6.2. Age

Each woman’s age in years was based on the survey return date.

#### 2.6.3. Weight and BMI

Women were asked to record their weight without clothes or shoes. BMI was calculated from self-reported height and weight by dividing each participant’s weight in kilograms by height in meters squared. We used World Health Organization-defined categories to classify women as underweight (BMI < 18.5 kg/m^2^), healthy weight (BMI 18.5 to 24.9 kg/m^2^), overweight (BMI 25.0 to 29.9 kg/m^2^), and obese (BMI ≥ 30.0 kg/m^2^) [42]. 

#### 2.6.4. Income

Women were asked to report their personal average individual gross (before tax) income, including pensions, allowances, and financial support from parents. Responses were categorized as no income and low (>0 to 36,399), medium (36,400 to 77,999), and high (>77,999) Australian dollars per annum [33]. 

#### 2.6.5. Education 

Women were asked to report their highest educational attainment, categorized into formal education/high school, trade/diploma, or degree or higher. 

#### 2.6.6. Marital Status 

Women were asked to indicate their present marital status. Responses were categorized into two categories of married/defacto or not married. 

#### 2.6.7. Occupation

Women were asked what their main occupation was at the time of the survey. Responses were amalgamated and categorized into four categories of no paid job, clerical/trade, associate professional, and professional [43]. 

#### 2.6.8. Employment at the Time of Childbirth

Women were asked about their participation in employment and payment for work. These responses were amalgamated and categorized as unemployed, volunteer, and employed.

#### 2.6.9. Maternity Leave Type (Paid/Unpaid)

Women were asked whether their maternity leave was paid (yes, no), with reference to the birth of their last child. 

#### 2.6.10. Duration of Maternity Leave

Women were asked the question “If you went back to paid work after the birth of your last child, how soon did you go back?”, which was answered in number of months. The responses were used to determine the duration of each woman’s maternity leave period as a continuous variable measured in months. 

#### 2.6.11. Heath Provider Access 

Women were asked to rate their access to maternal and child health services. Reponses were categorized as fair/poor, good, and very good/excellent. I don’t know responses were recoded as missing values. Women were also asked to record the number of times they had consulted a specialist doctor for their own health in the last 12 months. Responses were categorized as none, 1–4 times, 5–9 times, and 10–12 times.

#### 2.6.12. Breastfeeding

Women were asked to indicate their current breastfeeding status (no, yes).

#### 2.6.13. Smoking 

Women were asked about their smoking status, categorized into two broad categories, smoker and non-smoker. 

#### 2.6.14. Vitamin/Mineral Supplement Use

Women were asked to indicate how often they took vitamin/mineral supplements for their own health in the last 12 months. Responses were categorized as never, rarely, sometimes, and often.

#### 2.6.15. Depression and Anxiety 

Women were asked to indicate whether they had been diagnosed and treated for depression or anxiety disorder in the last 3 years. The responses for each of these conditions were amalgamated to no and no, no and yes, yes and no, and yes and yes, respectively. 

#### 2.6.16. Polycystic Ovary Syndrome (PCOS) 

Women were asked to indicate whether they had been diagnosed and treated for PCOS in the last 3 years. 

#### 2.6.17. Self-Rated Health

Women were asked to rate their overall health in general. Responses were categorized as fair/poor, good, very good, and excellent. 

### 2.7. Outcome and Data Analysis 

Differences in demographic variables, DGI and its components, energy, macronutrients, and micronutrient intake between groups of the study population, early (0–6 months postpartum), late (7–12 months postpartum), and all other women with children (>12 months after the birth of the last child), were analyzed using one way analysis of variance (ANOVA) to compare continuous variables or chi-square tests for categorical variables. Continuous variables were summarized as the mean with standard deviation and categorical variables as the number and percentage. Multiple linear regression analyses were performed to assess factors associated with DGI. The final model was constructed by including potential covariates associated with DGI in univariable linear regression analysis with (*p* ≤ 0.1) and using backwards elimination to remove variables no longer associated with DGI at *p* < 0.05. All regression models included age and BMI as clinically relevant variables associated with total diet quality [19,44]. All regression analyses were survey weighted to control for the oversampling of rural and remote women in the study population. All *p*-values were calculated from two-tailed tests of statistical significance with a type 1 error rate of 5%. All analyses were performed using Stata software version 15 (TX, USA [45]).

## 3. Results

### 3.1. Participant Characteristics

At survey 5, of the *n* = 4539 women in the total sample 12.3% (*n* = 558) of the women were categorized as early (0–6 months postpartum), 12.1% (*n* = 547) were categorized as late (7–12 months postpartum), and 75.7% (*n* = 3434) were categorized as all other women >12 months post childbirth. The demographic characteristics according to each category are reported in Table 1. Women >12 months post childbirth were older and more likely to be smokers, not married, employed, on a low income, and have two or more children in comparison to postpartum women. Postpartum women had higher educational attainment and were more likely to be in professional roles or in no paid job and unemployed, yet were more likely to be on a high income in comparison to women >12 months post childbirth. Postpartum women were more likely to be breastfeeding and taking vitamin and mineral supplements often. Women early postpartum were less physically active in comparison to women >12 months post childbirth.

### 3.2. Dietary Intake

The dietary intake categorized by the time since childbirth is reported in Table 2. Postpartum women had higher intake of energy, macronutrients (protein, fat, saturated fat, monounsaturated fat, polyunsaturated fat, and fiber) and micronutrients (iron, folate, sodium, zinc, magnesium, phosphorus, potassium, niacin, retinol, riboflavin, thiamine, and vitamin E) compared with women >12 months post childbirth. In comparison with women late postpartum and women >12 months post childbirth, women early postpartum had consumed a lower proportion of energy as protein and a higher proportion as carbohydrate intake. Women early postpartum had consumed a lower proportion of energy as fat compared with women >12 months post childbirth. 

Carbohydrate (g/day), glycemic load, calcium, and vitamin C intake were higher in women early postpartum compared with women late postpartum and also higher in women late postpartum compared with women >12 months post childbirth. Conversely, alcohol intake was highest in women >12 months post childbirth compared with women late postpartum and in women late postpartum compared with women early postpartum. 

Women late postpartum had a significantly higher cholesterol and beta-carotene intake and a lower glycemic index (GI) diet in comparison with women >12 months post childbirth. 

On adjustment for energy intake, these findings were maintained with the exception of higher potassium intake for postpartum women compared to women >12 months post childbirth (data not reported). 

### 3.3. Total Diet Quality and Its Components Categorized by Their Time since Childbirth

Total DGI and its components categorized by the time since childbirth are reported in Table 3. Postpartum women had a higher total DGI score than women >12 months post childbirth. This was supported by higher scores in diet variety, breads and cereals, proportion of wholegrains, and dairy and a lower saturated fat score (measured by the sum of the DGI dairy score and the DGI lean protein score divided by 2) in comparison to women >12 months post childbirth. 

Women late postpartum had a higher vegetable score in comparison to women early postpartum and women >12 months post childbirth. Women late postpartum scored higher in the lean meat and the proportion of lean meat relative to all meats consumed, and were more likely to consume reduced fat milk and less likely to consume full cream milk, soya, or skimmed milk in comparison to women >12 months post childbirth. 

Women early postpartum were more likely to exceed recommendations for extra foods, scoring greater than 2.5 serves of extra foods per day in comparison to women >12 months post childbirth. 

### 3.4. Contributors of Demographic and Maternity-Related Factors to Total Diet Quality 

The independent associations of anthropometric, demographic, and maternity-related factors with DGI in women are reported in Table 4. In unadjusted analyses education, physical activity, access to maternal and child health services, vitamin/mineral supplement use, paid maternity leave verses no paid maternity leave, breastfeeding versus currently not breastfeeding, frequency of being seen by a specialist doctor, and self-rated health were positively associated with DGI. Women with a higher DGI included those with the shortest time since childbirth (<12 months) compared to women >12 months since childbirth, women who were unemployed or volunteers compared to employed women, and women in professional roles in comparison to women in no paid job. Women with a lower DGI included smokers versus non-smokers, underweight, overweight, or obese women, in comparison to normal weight women, women on a low and medium personal income compared to women on no income, women in associate professional and clerical trade roles in comparison to women in no paid job, unmarried women in comparison to married women, and women with depression and anxiety, in comparison to women without depression and anxiety.

In multivariable linear regression analysis education, physical activity, better access to maternal and child health services, and vitamin and mineral supplement use were positively associated with DGI, and increasing time from childbirth (>12 months versus <12 months), smoking, and medium income compared to no income level were negatively associated with DGI. Other demographic and maternal factors were not predictive of DGI in the adjusted analysis. 

## 4. Discussion

In a large population-based cohort of reproductive-aged women with children, using detailed dietary assessment, we reported that women within 12 months of childbirth had better diet quality, as indicated by total DGI and its components, including a higher diet variety, higher intake of breads and cereals, and a higher proportion of wholegrains and dairy intake in comparison to women who had given birth more than 12 months ago. We also report a higher intake of energy, protein, fat, fiber, and micronutrients in postpartum women compared with women >12 months post childbirth. Factors positively associated with a higher diet quality in women were higher educational attainment, frequent vitamin/mineral supplement use, physical activity, and greater access to maternal and child health services. Conversely, increasing time from childbirth (>12 months), medium income level, and smoking were negatively associated with diet quality in women. Our finding of a reduced diet quality in women more than 12 months after childbirth compared with those in the first 12 months postpartum may be explained by differences in family dynamics. Women who were not within 12 months postpartum were more likely to report having had more children and were more likely to be engaged in the workforce. We were unable to account for children in our final model due to collinearity with time since childbirth. Previous research has shown that employed women with two or more dependent children report a higher prevalence of work-life conflict in comparison to employed women with one or no dependent children [46]. Increased work-life conflict has been associated with a greater likelihood of negative mental health consequences, including worsened self-rated health, negative emotions and depression, low energy and optimism, and regular fatigue [46]. Poor self-rated health and depressive symptoms have been associated with a lower diet quality in women [47]. Time-related barriers, such as irregular working hours, a busy lifestyle, and the food preferences of children, are also shown to be associated with a reduced frequency of consumption of vegetables and home cooked meals in women with children [48,49]. These barriers to healthy eating may potentially contribute to the reduced diet variety and lower diet quality overall, as found here in women >12 months post childbirth. Given that women are often the main influencer for family dietary intake and the main food preparers in the home [50], this may translate to unfavorable dietary habits in children, as previously shown [24]. 

Postpartum women were more likely to be on maternity leave, reflective of increased reporting of unemployment, and not earning any personal income. The negative association between medium income level and diet quality is unexpected as higher income is generally associated with improved diet quality [19]. However, we did not see any association between the highest income category and diet quality. We note wide confidence intervals and the small number of women in this group. Additionally, women may have been supported by family income or additional entitlements (government support for example) during maternity leave, which may act to offset financial burden and confounding paid/unpaid maternity leave status during this time. We are unable to be conclusive, since information on government entitlements and family income were not collected. We acknowledge this as a limitation to the study. 

We note subtle differences in diet quality between the early to late postpartum women. Women late postpartum had a lower GI diet, compared to women >12 months post childbirth, and better diet quality, relating to higher vegetable intake, compared to early postpartum women and women >12 months post childbirth. Early postpartum women had higher carbohydrate (g/day), glycemic load, calcium, and vitamin C intake compared to women in late postpartum and women >12 months post childbirth. In comparison with women late postpartum, early postpartum women may be experiencing greater disruptions to sleep [51], stress, and depressive symptoms [52], of which the latter have been associated with reduced postpartum diet quality [22], consistent with our result of a lower diet quality in women with depression and anxiety, in comparison to women without depression and anxiety. Depressive symptoms may be a contributor to the differences in diet quality seen here between the early and late postpartum women. 

Women within 12 months of childbirth reported better access to maternal and child health services (MCH) than women more than 12 months after childbirth, which in turn may provide more social support. Increased social support has previously been shown to be associated with a higher diet quality in mothers [53], consistent with our result of a positive independent association between diet quality and access to MCH. In addition, increased emphasis on child and family nutrition by MCH, typically coinciding with introduction to solids at approximately 6 months postpartum [54], may contribute to improved diet quality postpartum. Research also shows that focusing on parenting self-efficacy is effective in improving dietary and physical activity behaviors in school-aged overweight and obese children [55]. Therefore, parenting support to mothers beyond the postpartum period may provide an avenue to a healthier lifestyle for the whole family. 

Our result of a positive association between diet quality and a higher education and higher physical activity levels are consistent with the literature [56]. We also report that women with children who smoke are at risk of poorer diet quality, consistent with prior research of a negative association between diet quality and lower vegetable, fruit, wholegrain, and low fat milk intake and vitamin/mineral supplement use in smokers, independent of socioeconomic, lifestyle, and biological confounding factors [57,58]. Our findings of lower self-reported smoking rates and improved diet quality within the first year of childbirth may also reflect that women have given up smoking for the duration of pregnancy and improved dietary intake to benefit their infants. These results highlight the need for effective interventions to support women in the years following childbirth to prevent resumption of smoking. 

### 4.1. Strengths 

The strengths of this study include the large, unselected community cohort with high baseline and ongoing participation rates. The community-based nature of the sample minimizes selection bias to include women with a variety of clinical presentations. A comparison of women who participated in the baseline survey with data from women in the same age range from the Australian census of 1996 showed that the ALSWH participants were reasonably representative of the general population [28,30]. 

### 4.2. Limitations

The limitations of this study include reliance on self-reported data, although this is common to most large prospective community-based cohort studies over multiple time points. However, self-reported BMI has been validated and has been found to correlate well with measured BMI in another age cohort of the ALSWH population [59] and is considered appropriate in studies involving younger adults in large epidemiological studies [60]. Self-report FFQs can result in misreporting, recall error, measurement error, and induce social desirability bias [61]. To minimize these errors, we used an FFQ validated against weighed food records within an Australian population, which has been shown to provide a useful method for measuring habitual dietary intake in population settings [34,62]. FFQs are generally acceptable as a main method of dietary intake in studies of this type and size [61]. Some women may have reported a dietary recall when they were pregnant, which may be different to their dietary intake postpartum; however, evidence suggests that there are minimal dietary differences between pregnancy and postpartum [25,63]. Additionally, while we attempted to broadly explore predictors of diet quality in this population, we cannot rule out other confounding factors that may account for differences between these groups that we have not accounted for in our model, such as psychosocial factors, including weight-related distress and perceived barriers to weight loss and body image [22], all of which have been previously shown to be associated with reduced adherence to nutrition recommendations. Similarly, we explored the association between diet quality and the number of children; however, due to collinearity with time since childbirth, the number of children variables did not fit the final model and was therefore removed. However, we acknowledge that this variable may be a confounder affecting the final outcome. There may have been differences in the interpretation of the maternity leave and employment questions, as some women on maternity leave may consider themselves as currently unemployed; therefore, this is a limitation in the accuracy of the data in the employment and maternity leave variables. 

## 5. Conclusions

Our main finding of a reduced diet quality associated with time since childbirth may be reflective of common factors observed in women beyond the postpartum period, including work-life conflict, time related barriers, and family food preferences. Given that weight, obesity, and cardiovascular risk factors are increasing in women, and improved diet quality lowers disease risk overall, this study highlights the need to support women beyond the postpartum period to improve overall diet quality and associated health behaviors, including smoking, for reduced chronic disease risk. 

## Figures and Tables

**Table 1 jcm-09-00446-t001:** Characteristics of women categorized by their time since childbirth.

Characteristic	Early Postpartum	Late Postpartum	Women >12 Months Post Childbirth	*p*-Value
	(0–6 Months)	(7–12 Months)	*n* = 3434	
	*n* = 558	*n* = 547		
Age (years)	33.6 (1.4)	33.6 (1.4)	34.0 (1.4)	<0.001 *
Smoking status	519 (93.2)	499 (91.4)	2854 (83.2)	<0.001 *
Non-smoker	38 (6.8)	47 (8.6)	575 (16.8)	
Smoker	71.2 (15.3)	70.7 (15.1)	71.7 (16.7)	
Weight (kg)	25.9 (5.2)	25.6 (2.3)	26.2 (5.9)	0.386
BMI (kg/m^2^)				0.023 ****
Underweight (<18.5)	4 (0.73)	12 (2.2)	96 (2.9)	
Normal (18.5–24.9 kg/m^2^)	280 (50.9)	286 (53.1)	1647 (49.1)	
Overweight (25–29.9 kg/m^2^)	164 (29.8)	136 (25.2)	900 (26.8)	
Obese (≥30 kg/m^2^)	102 (18.6)	105 (19.5)	710 (21.2)	
Country of birth				0.206
Australian born	528 (95.3)	503 (92.8)	3198 (93.7)	
Overseas born	26 (4.7)	39 (7.2)	216 (6.3)	
Personal annual income				<0.001 **
No income	97 (19.5)	99 (19.8)	341 (11.0)	
Low ($AUD > 0–36,399)	217 (43.7)	256 (51.3)	1786 (57.4)	
Medium ($AUD 36,400–77,999)	138 (27.8)	96 (19.2)	783 (25.1)	
High ($AUD > 77,999)	45 (9.1)	48 (9.6)	204 (6.6)	
Education				<0.001 *
No formal /high school	78 (14.2)	93 (17.2)	984 (29.5)	
Trade/diploma	126 (23)	133 (24.6)	1018 (30.5)	
Degree	344 (62.8)	315 (58.2)	1333 (40)	
Occupation				<0.001 *
No paid job	231 (41.9)	220 (40.7)	849 (25.4)	
Clerical trade	32 (5.8)	48 (8.9)	799 (23.9)	
Assoc. professional	63 (11.4)	73 (13.5)	588 (17.6)	
Professional	225 (40.8)	199 (36.9)	1113 (33.2)	
Employment				<0.001 **
Unemployed	331 (59.3)	244 (44.7)	722 (21.1)	
Volunteer	55 (9.9)	70 (12.8)	295 (8.6)	
Employed	172 (30.8)	232 (42.5)	2403 (70.3)	
Marital status				<0.001 *
Married/defacto	546 (98)	529 (96.7)	2990 (87.4)	
Not married	11 (2)	18 (3.3)	433 (12.7)	
Breastfeeding				<0.001 **
Not currently breastfeeding	428 (76.8)	462 (84.8)	3397 (99)	
Currently breastfeeding	129 (23.2)	83 (15.2)	35 (1)	
Number of children				<0.001 *
1	214 (38.4)	201 (36.8)	841 (24.5)	
2	344 (61.6)	346 (63.3)	2593 (75.5)	
Physical activity (MetMin)	555.1 (668.4)	669.8 (731.3)	795.4 (1039.2)	<0.001 ****
Vitamin/mineral supplement use				<0.001 **
Never	27 (4.8)	40 (7.3)	707 (20.6)	
Rarely	31 (5.6)	68 (12.4)	631 (18.4)	
Sometimes	110 (19.7)	143 (26.1)	930 (27.1)	
Often	390 (69.9)	296 (54.1)	1161 (33.9)	

Data are presented as mean (standard deviation) for continuous variables and frequency and percentage *(%)* for categorical variables. Data were analyzed by ANOVA for continuous variables (age, physical activity, weight, body mass index (BMI)) and chi square tests for categorical variables. Body mass index (BMI); metabolic equivalent value (MetMin). * Statistically significant difference (*p* < 0.05) between early postpartum compared to women >12 months post childbirth and late postpartum compared to women > 12 months post childbirth. ** Statistically significant difference (*p* < 0.05) between all three groups. *** Statistically significant difference (*p* < 0.05) between early postpartum compared to late postpartum and early postpartum compared to women > 12 months post childbirth. **** Statistically significant difference (*p* < 0.05) between early postpartum compared to women >12 months post childbirth.

**Table 2 jcm-09-00446-t002:** Energy, macronutrient, and micronutrient intake in women categorized by their time since childbirth.

	Early Postpartum (0–6 Months) *n* = 558	Late Postpartum (7–12 Months) *n* = 547	Women >12 Months post Childbirth *n* = 3430	*p*-Value
Energy (KJ/day)	7879.5 (2357.6)	7720.7 (2294.8)	7088.0 (2304.0)	<0.001 *
Energy (Kcal)	1883.2 (563.5)	1845.3 (548.5)	1694.1 (550.7)	<0.001 *
Protein (g/day)	90.1 (27.7)	91.5 (29)	83.9 (28.8)	<0.001 *
Protein (% energy)	19.9 (3.0)	20.8 (3.1)	21.1 (3.4)	<0.001 **
Carbohydrate (g/day)	201.8 (65.4)	189.2 (59.5)	169.5 (59.4)	<0.001 ***
Carbohydrate (% energy)	41.6 (4.8)	40.1 (4.7)	39.6 (5.6)	<0.001 **
Fat (g/day)	77.5 (26.7)	76.5 (26.3)	69.8 (26.6)	<0.001 *
Fat (% energy)	36.7 (4.6)	37.3 (4.1)	37.6 (4.9)	0.0003 ****
Saturated fat (g/day)	33.2 (12.6)	32.4 (12.1)	29.6 (12.2)	<0.001 *
Saturated fat (% energy)	15.7 (3.1)	15.8 (2.7)	15.9 (3.0)	0.409
Monounsaturated fat (g/day)	27.1 (9.6)	27.1 (9.6)	24.8 (9.7)	<0.001 *
Monounsaturated fat (% energy)	12.6 (1.8)	12.9 (1.8)	12.8 (2.0)	0.097
Polyunsaturated fat (g/day)	10.4 (4.7)	10.4 (4.5)	9.3 (4.3)	<0.001 *
Polyunsaturated fat (% energy)	4.9 (1.6)	5.0 (1.5)	4.8 (1.5)	0.062
Alcohol (g/day)	3.7 (6.6)	5.9 (9.7)	8.6 (12.8)	<0.001 ***
Fiber (g/day)	21.7 (7.8)	21.2 (7.1)	18.6 (6.8)	<0.001 *
Cholesterol (mg/day)	282.0 (106.5)	289.7 (107.7)	273.0 (110.9)	0.002 *****
Glycemic index	51.2 (3.7)	50.9 (3.7)	51.4 (4.1)	0.031 *****
Glycemic load	103.4 (35.9)	96.5 (33.0)	87.1 (33.5)	<0.001 ***
Calcium (mg/day)	990.1 (314.8)	946.2 (296.1)	842.3 (281.0)	<0.001 ***
Iron (mg/day)	13.5 (5.0)	13.4 (4.9)	11.7 (4.5)	<0.001 *
Folate (µg/day)	281.8 (96.3)	275.6 (92.6)	239.6 (84.7)	<0.001 *
Sodium (mg/day)	2514.8 (820.0)	2503.9 (832.1)	2305.5 (838.0)	<0.001 *
Zinc (mg/day)	11.9 (3.8)	12.0 (3.9)	11.0 (4.0)	<0.001 *
Magnesium (mg/day)	303.4 (96.2)	296.2 (89.8)	260.7 (84.4)	<0.001 *
Phosphorus (mg/day)	1601.1 (477.6)	1576.7 (457.9)	1414.1 (444.9)	<0.001 *
Potassium (mg/day)	2885.6 (801.4)	2841.9 (795.8)	2564.6 (755.4)	<0.001 *
Beta-carotene (µg/day)	2648.9 (1234.9)	2821.5 (1369.2)	2572.9 (1227.3)	<0.001 *****
Niacin (mg/day)	22.3 (8.2)	22.4 (8.2)	19.5 (7.6)	<0.001 *
Retinol (µg/day)	371.6 (155.3)	357.8 (149.5)	318.3 (145.4)	<0.001 *
Riboflavin (mg/day)	2.6 (0.94)	2.6 (0.89)	2.2 (0.83)	<0.001 *
Thiamin (mg/day)	1.6 (0.63)	1.6 (0.61)	1.3 (0.56)	<0.001 *
Vitamin C (mg/day)	121.2 (64.0)	108.8 (57.8)	101.5 (52.5)	<0.001 ***
Vitamin E (mg/day)	6.3 (2.3)	6.2 (2.2)	5.5 (2.1)	<0.001 *

Data are presented as mean (standard deviation). Data were analyzed by ANOVA. * Statistically significant difference (*p* < 0.05) between early postpartum compared to women >12 months post childbirth and late postpartum compared to women >12 months post childbirth. ** Statistically significant difference (*p* < 0.05) between early postpartum compared to late postpartum and early postpartum compared to women >12 months post childbirth. *** Statistically significant difference (*p* < 0.05) between all three groups. **** Statistically significant difference (*p* < 0.05) between early postpartum compared to women >12 months post childbirth. ***** Statistically significant difference (*p* < 0.05) between late postpartum compared to women >12 months post childbirth.

**Table 3 jcm-09-00446-t003:** Total diet quality and its components in women categorized by their time since childbirth.

Diet Quality Component	Early Postpartum (0–6 Months) *n* = 558	Late Postpartum (7–12 Months) *n* = 547	Women > 12 Months Post Childbirth *n* = 3434	*p*-Value
Diet variety	5.7 (1.2)	5.6 (1.3)	5.2 (1.4)	<0.001 *
Vegetables	4.3 (1.8)	4.6 (1.9)	4.4 (1.9)	0.002 **
Fruit	10.0 (0.43)	9.9 (0.65)	9.9 (0.79)	0.429
Breads and cereals	6.1 (2.3)	5.9 (2.2)	5.0 (2.3)	<0.001 *
Wholegrain proportion	7.6 (4.2)	7.9 (4.1)	6.7 (4.7)	<0.001 *
Lean meat	9.6 (1.1)	9.7 (1.1)	9.5 (1.3)	0.011 ***
Lean meat proportion	8.3 (1.0)	8.4 (1.0)	8.2 (1.0)	0.009 ***
Dairy	8.5 (1.9)	8.3 (2.0)	7.6 (2.3)	<0.001 *
Type of milk consumed				0.024 ***
No milk, soya, skim milk	165 (29.6)	152 (27.8)	1088 (31.7)	
Reduced fat milk	206 (36.9)	205 (37.5)	1094 (31.9)	
Full cream milk	187 (33.5)	190 (34.7)	1249 (36.4)	
Saturated fat	8.4 (1.1)	8.3 (1.1)	7.9 (1.2)	<0.001 *
Extra foods				0.043 ****
≤2.5 serves/day)	4 (0.72)	9 (1.7)	78 (2.3)	
>2.5 serves per day	554 (99.3)	538 (98.4)	3356 (97.7)	
Total DGI	89.8 (10.5)	90.0 (10.2)	85.2 (11.7)	<0.001 *

Data are presented as mean (standard deviation) for continuous variables and frequency and percentage (%) for categorical variables. Data were analyzed by ANOVA and the chi-squared test for categorical outcomes. Dietary guideline index (DGI). * Statistically significant difference (*p* < 0.05) between early postpartum compared to women >12 months post childbirth and late postpartum compared to women >12 months post childbirth. ** Statistically significant difference (*p* < 0.05) between early postpartum compared to late postpartum and late postpartum compared to women >12 months post childbirth. *** Statistically significant difference (*p* < 0.05) between late postpartum compared to women >12 months post childbirth. **** Statistically significant difference (*p* < 0.05) between early postpartum and women >12 months post childbirth.

**Table 4 jcm-09-00446-t004:** Contributors of demographic and anthropometric factors to total diet quality in all women.

Characteristics	Unadjusted β (95% CI)	*p*-Value	Adjusted β (95% CI)	*p*-Value
Age (years)	−0.14 (−0.34, 0.09)	0.232	0.05 (−0.23, 0.32)	0.742
Smoking status				
Non-smoker	Ref		Ref	
Smoker	−7.0 (−7.9, −6.1)	<0.001	−4.1 (−5.4, −2.8)	<0.001
BMI (kg/m^2^)				
Underweight (<18.5)	−2.13 (−4.3, 0.05)	0.056	−1.0 (−3.6, 1.6)	0.455
Normal weight (18.5–24.9)	Ref		Ref	Ref
Overweight (25–29.9)	−0.61 (−1.4, 0.20)	0.138	−0.07 (−1.0, 0.86)	0.882
Obese (BMI ≥ 30)	−1.8 (−2.7, −0.93)	<0.001	0.08 (−1.1, 1.2)	0.895
Education				
No formal /high school	Ref		Ref	
Trade/diploma	2.2 (1.3, 3.1)	<0.001	1.7 (0.49, 3.0)	0.006
Degree	6.2 (5.4, 7.0)	<0.001	4.4 (3.3, 5.5)	<0.001
Personal annual income				
No income	Ref		Ref	
Low ($AUD > 0–36,399)	−2.3 (−3.4, −1.2)	<0.001	−0.85 (−2.0, 0.30)	0.148
Medium ($AUD 36,400–77,999)	−2.5 (−3.7, −1.3)	<0.001	−2.0 (−3.3, −0.74)	0.002
High ($AUD > 77,999)	−0.79 (−2.4, 0.82)	0.335	−1.1 (−2.8, 0.57)	0.2
Employment		<0.001	N/A	0.964
Unemployed	1.9 (1.2, 2.7)	0.032		
Volunteer	1.3 (0.11, 2.5)			
Employed	Ref			
Occupation			N/A	0.124
No paid job	Ref			
Clerical trade	−3.3 (−4.3, −2.3)	<0.001		
Associate professional	−2.1 (−3.1, −1.1)	<0.001		
Professional	0.94 (0.09, 1.8)	0.03		
Marital status				
Married/defacto	Ref		Ref	
Not married	−4.1 (−5.2, −3.0)	<0.001	−1.6 (−3.4, 0.11)	0.067
Physical activity (MetMin)	0.002 (0.001, 0.002)	<0.001	0.001 (0.0007, 0.002)	<0.001
Maternal and child health access				
Fair/poor	Ref		Ref	
Good	1.2 (0.008, 2.4)	0.048	1.1 (−0.35, 2.5)	0.141
Excellent/very good	2.3 (1.2, 3.4)	<0.001	1.4 (0.10, 2.6)	0.034
Vitamin/mineral supplement use				
Never	Ref		Ref	
Rarely	0.96 (−0.19, 2.10)	0.103	0.96 (−0.48, 2.4)	0.192
Sometimes	2.1 (1.0, 3.1)	<0.001	1.8 (0.47, 3.1)	0.008
Often	5.3 (4.3, 6.2)	<0.001	3.9 (2.6, 5.1)	<0.001
Time since last childbirth				
0–6 months	4.6 (3.6, 5.7)	<0.001	2.0 (0.79, 3.1)	0.001
7–12 months	4.8 (3.8, 5.8)	<0.001	2.4 (1.3, 3.5)	<0.001
Women >12 months post childbirth	Ref		Ref	
Depression/anxiety in the last 3 years			N/A	0.749
No and no	Ref			
No and yes	−0.85 (−2.9, 1.2)	0.417		
Yes and no	−2.4 (−3.5, −1.3)	<0.001		
Yes and yes	−1.7 (−3.2, −0.15)	0.032		
PCOS in the last 3 years			N/A	N/A
No	Ref			
Yes	1.0 (−0.60, 2.7)	0.213		
Paid maternity leave status			N/A	0.156
No	Ref			
Yes	3.0 (2.3, 3.8)	<0.001		
Duration of maternity leave (months)	−0.02 (−0.06, 0.01)	0.205	N/A	N/A
Specialist Doctor			N/A	0.633
None	Ref			
1–4 times	1.2 (0.40, 1.9)	0.003		
5–9 times	3.0 (1.8, 4.3)	<0.001		
10–12 times	4.4 (3.2, 5.6)	<0.001		
Breastfeeding			N/A	0.188
Not currently breastfeeding	Ref			
Currently breastfeeding	4.5 (3.0, 6.0)	<0.001		
Self-rated health			N/A	0.056
Fair/poor	Ref			
Good	2.4 (1.1, 3.6)	<0.001		
Very good	4.2 (3.0, 5.4)	<0.001		
Excellent	5.8 (4.4, 7.2)	<0.001		

Data are presented as beta, 95% confidence interval, and *p*-value and were analyzed using survey weighted multiple linear regression. Body mass index (BMI); metabolic equivalent value (MetMin) polycystic ovary syndrome (PCOS); variable not included in the final model due to insignificant univariate association or was removed via backwards elimination (N/A).

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
