# Peer review of "Postpartum Diet Quality: A Cross-Sectional Analysis from the Australian Longitudinal Study on Women’s Health"

_jcm, 2020, doi:10.3390/jcm9020446_

Round 1

Reviewer 1 Report

I have a few comments on issues that should be addressed before the final decision on this paper:

The authors repeatedly use the expression “retention of baseline participants” (both in the Abstract section and in the main text). Is it referring to the main “survey 5”, as it would seem, or is it rather pointing to the procedure the authors used to select the final group of subjects? In the case of the former, as very few details about the structure of the initial survey are provided, we have no explanation for a rather low retention rate, and more information might prove useful. In the case of the latter, we are not discussing a true retention rate, since this is rather the direct result of the exclusion criteria applied to select suitable participants from a larger cohort, so the expression should be changed to avoid confusion. Could these results be extrapolated to other populations, or are they to be interpreted in the context of their peculiar geographical and cultural setting? Is it possible to trace any comparison between the data obtained by the authors and the scarcely existing data on other populations outside Australian territory? No matter if the answer to these questions is positive or negative, related commentaries may help to increase the international relevance of these data and may deserve a place among the Discussions (or maybe the limits) of the paper. It appears that the results of the multiple linear regression analysis are not accompanied by enough discussion in the dedicated section of the paper. If possible, the Discussion section should be extended to provide more comments related to this very interesting set of results.

Author Response

Dear Reviewer 1, 

Thank you for your feedback regarding this manuscript and the opportunity to amend the paper. Please see the attachment. 

Thank you,

Julie Martin

Reviewer 2 Report

The authors analyze the association between time since childbirth with diet quality among females aged 31-36 at time of childbirth.

Major comments

The reason for the division of the study population into early (0-6 m) and late (7-12 m) post-partum and >12 m was not clear, and should appear in the methods section. The 2 first groups (1-6 and 7-12 m) differ in some categories, but are similar with respect to diet quality and food intake. The authors also refer to these 2 groups as one in some instances. The authors may choose to combine the 2 post-partum groups, and discuss the early (<12 m) vs late (>12 m) differences.

I did not see whether number of children was added as an independent parameter in the regression model. It would be interesting to see whether the number of children is a factor influencing the quality of diet in this population.\

Minor comments:

The authors did not add number of children as a confounder, though added in the tables.

The authors did not address the fact that caloric intake in the 12 months following birth was higher compared to >12 months women, despite the fact the BMI was slightly higher among >12 months sub-group.

I would ask the authors to add the Kcal values as well.

The manuscript contains a large amount of data. The amount of data is sometimes hard to follow. I would recommend editing to enable easier reading

The list of confounders is long, and many of the variables were listed only as self-reported. The authors should concentrate the list of confounders in one paragraph. Important information about specific parameters (like weight and BMI) should be added afterwards. Sections 3.2 and 3.3 are cumbersome. The text would be easier to read if the 2 post-partum groups were used as 1, and compared to the >12 months group. This is also appropriate since most of the data is similar between the 2 post-partum groups. The tables are more than enough to understand the differences between the 3 sub-groups. Section 3.4 – please re-phrase so it would be easier to read.

Author Response

Dear Reviewer 2, 

Thank you for your feedback regarding the attached manuscript and the opportunity to amend this paper. Please refer to the attachment. 

Regards,

Julie Martin 

Round 2

Reviewer 2 Report

No additional comments